# SHAPING ROBOTIC ACTIONS WITH FOURIER FLOW MATCHING

## ABSTRACT

We present a Fourier-based flow-matching method for Vision-Language-Action (VLA) policies that lets the policy reason over smooth trajectories, rather than stepwise actions. Instead of training on raw joint- or Cartesian-space action sequences, we project each sequence into a compact Discrete Cosine Transform (DCT) basis and learn directly in coefficient space via flow matching. This trajectory-level representation enforces smoothness and reduces dimensionality. Importantly, we show that the DCT representation integrates with asynchronous plan-execute schemes, preserving policy responsiveness. In experiments, predicting DCT coefficients yields higher task success than classical flow matching VLA baselines trained on per-step actions. Our results indicate that Fourier-domain flow matching is a simple, drop-in alternative that improves the performance and stability of VLA policies.

## 1 INTRODUCTION

Vision–Language–Action (VLA) policies extend large vision–language models (VLM) with robot-executable actions, grounding perception and language in real-world control (Team et al., 2024; Kim et al., 2024; Black et al., 2024). While architectures differ, most prior VLA work predicts actions stepwise in joint or Cartesian space, implicitly treating demonstrations as discrete sequences rather than samples from an underlying continuous trajectory. This per-step view can inject noise and discontinuities and overlooks global structure that could simplify learning and improve accuracy.

Flow matching (Lipman et al., 2023) has recently been adopted for VLAs (Black et al., 2024), learning a time-conditioned vector field that transports noise into task-appropriate actions given the current observations. In typical formulations, however, the learned flow acts directly on raw action sequences and therefore inherits the limitations of stepwise representations.

We propose Fourier Flow Matching (FFM), which performs flow matching in a compact Fourier domain. Each action trajectory is projected into a low-dimensional set of Discrete Cosine Transform (DCT) coefficients (Ahmed et al., 1974), and the model learns a velocity field over this coefficient space; trajectories are reconstructed by inverse DCT at inference. This trajectory-level representation enforces smoothness, reduces dimensionality, and lets the policy reason over continuous signals rather than discrete steps. Crucially, operating in coefficient space decouples planning from the control rate and integrates cleanly with asynchronous plan–execute schemes, preserving responsiveness.

Across simulated and real-robot evaluations, predicting DCT coefficients with FFM improves success rates and completion time over a flow-matching baseline trained on per-step actions, while offering the above representational benefits.

## 2 RELATED WORK

### 2.1 VISION–LANGUAGE–ACTION MODELS

VLAs map visual observations, language goals, and robot state to actions via a VLM backbone, grounding high-level reasoning in control (Team et al., 2024; Kim et al., 2024; Black et al., 2024). Early systems generated actions autoregressively (Team et al., 2024; Brohan et al., 2023), showing broad generalization but operating on per-step action tokens that underutilize trajectory structure.

Subsequent work introduced action chunking to expose longer temporal dependencies, improving stability and performance (Kim et al., 2025; 2024). Unlike chunking—which still reasons over discrete time steps—our approach reasons directly over a continuous trajectory parameterization.

## 2.2 FLOW-MATCHING VLAS

Flow matching learns a time-conditioned vector field that transports a simple base distribution to the data distribution (Lipman et al., 2023). Applied to VLAs, conditioning the flow on a VLM backbone yields strong visuomotor policies: noisy action proposals are integrated into observation-consistent actions (Black et al., 2024). However, existing flow-matching VLAs typically operate in raw joint/Cartesian action space and remain stepwise, which can induce discontinuities and ignore inherent trajectory smoothness. In contrast, we train the flow in a low-dimensional Fourier space: the policy predicts DCT-II coefficients, and actions are recovered via inverse DCT with a controller-appropriate rate.

## 2.3 COMPACT SIGNAL REPRESENTATIONS

Compact parameterizations exploit trajectory structure to reduce dimensionality and ease learning. In autoregressive VLAs, recent work compresses action sequences with tokenizers based on Fourier-like transforms or splines, reporting faster training and higher accuracy (Pertsch et al., 2025; Zhou et al., 2025). Closest to our direction, these methods rely on learned codebooks (often large, separately trained "universal" tokenizers) and discrete token generation. By contrast, we learn a probability flow directly in continuous DCT-II coefficient space—no tokenizer or codebook pretraining—then reconstruct trajectories by inverse DCT at whatever control rate the low-level controller requires.

## 3 FLOW-MATCHING VLAS IN RAW ACTION SPACE

Black et al. (2024) apply flow matching directly to stepwise actions in joint/Cartesian space. For a planning horizon $N$ and $d$-DoF actions, the policy learns a time-conditioned vector field over actions $\mathbf{A} \in \mathbb{R}^{d \times N}$ that transports a base distribution to task-consistent plans given observations $\boldsymbol{o}$ (images, language, robot state). Concretely, the learned velocity field

$$\frac{d\mathbf{A}_\tau}{d\tau} \; = \; v_\theta(\mathbf{A}_\tau, \boldsymbol{o})$$

is trained with optimal transport Gaussian flow to align the induced probability path with the data distribution (Lipman et al., 2023). After integrating this field, a prefix of the plan ("action chunk") is dispatched for execution and updated as soon as the next chunk is ready.

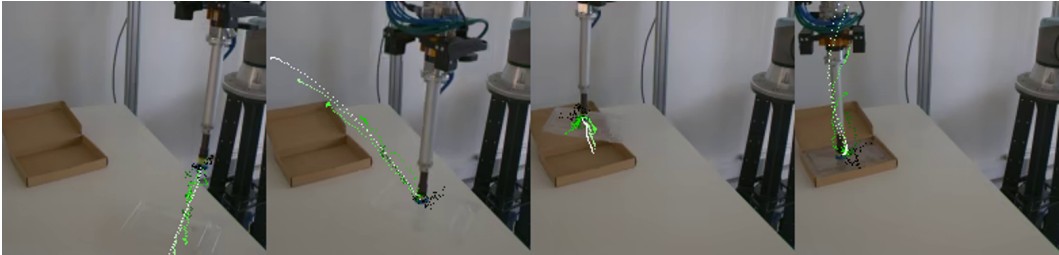

Figure 1: Flow matching VLAs turn noisy action samples (black) into progressively more precise actions (green to white) conditioned on current observations (images, text, and robot state). Early plans considered by the model have clear stepwise structure.

While effective, this representation treats demonstrations as per-step values rather than *samples from an underlying continuous trajectory*. Two practical issues follow. First, plans exhibit stepwise discontinuities, especially early in integration when $\mathbf{A}_t$ is still close to the noisy base (Figure 1).

Second, the model must devote capacity to explaining high-frequency artifacts introduced by discretization, rather than the lower-frequency task geometry; this also entangles the planning representation with the controller's execution rate, coupling learning dynamics to a system parameter that should be orthogonal to planning.

# 4 FOURIER FLOW MATCHING

We perform flow matching in the Fourier domain, where smooth trajectories admit a compact coefficient representation.

## 4.1 DCT REPRESENTATION

Each action sequence is encoded by a low-dimensional vector of Discrete Cosine Transform type-II (DCT-II) coefficients; the flow is learned over this coefficient space. For a 1-DoF sequence $x_{0:N-1}$,

$$f_k = c_k \sum_{n=0}^{N-1} x_n \cos\Big(\frac{\pi}{N}\big(n + \tfrac{1}{2}\big)k\Big), \qquad c_0 = \sqrt{\tfrac{1}{N}}, \ \ c_{k>0} = \sqrt{\tfrac{2}{N}}, \tag{1}$$

with inverse

$$x_n = \sum_{k=0}^{N-1} c_k f_k \cos\Big(\frac{\pi}{N}\big(n + \tfrac{1}{2}\big)k\Big). \tag{2}$$

For $d$-DoF actions $\boldsymbol{a}_n \in \mathbb{R}^d$, we apply equation 1 per dimension and retain only the first $K$ coefficients per dimension, yielding $\mathbf{F} \in \mathbb{R}^{d \times K}$ (a compact surrogate for the stepwise plan $\mathbf{A}$). We denote by $\text{IDCT}_N(\cdot)$ the inverse mapping that reconstructs a $d \times N$ action sequence from $\mathbf{F}$.

## 4.2 FLOW MATCHING OVER DCT COEFFICIENTS

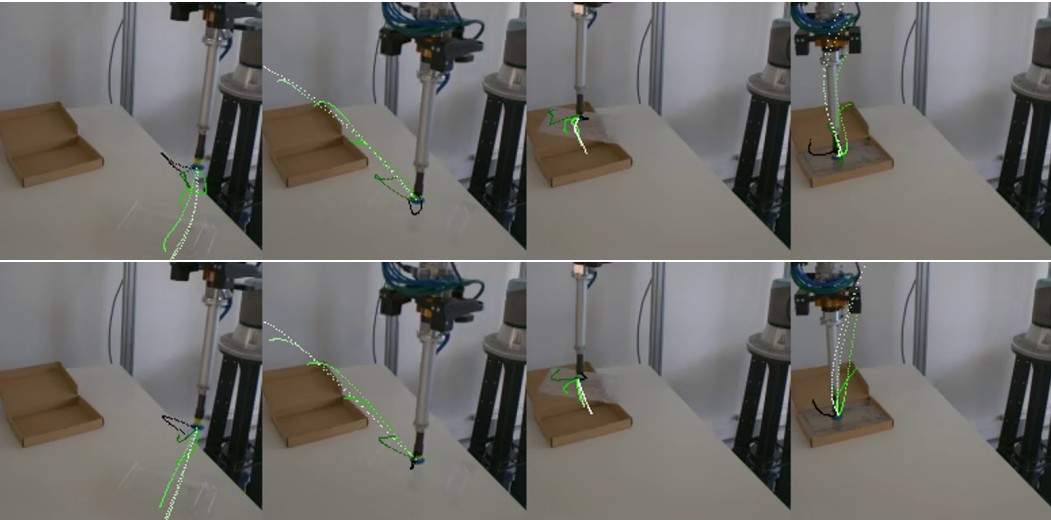

Figure 2: Fourier flow matching with $K = 5$ (top row) and $K = 2$ (bottom row). Inferences of different models are shown here on recorded observations from the same trajectory.

Let $\mathbf{F}_\tau$ denote time-dependent coefficients along the probability path from a simple base (e.g., $\mathcal{N}(0, \boldsymbol{I})$) to the data distribution, conditioned on observations $\mathbf{o}$ (images, language, state) and flow time $\tau \in [0, 1]$. We train a vector field $v_\theta(\mathbf{F}_\tau, \boldsymbol{o})$ with the standard flow-matching objective so that integrating

$$\frac{d\mathbf{F}_\tau}{d\tau} = v_\theta(\mathbf{F}_\tau, \boldsymbol{o})$$

transports noise to task-consistent coefficients. Operating in coefficient space makes the policy reason over *continuous* trajectories in a low-dimensional basis (Figure 2), reduces prediction dimensionality from $d \times N$ to $d \times K$, and decouples planning from the controller's sample rate.

### 4.3 INFERENCE AND CONTROL

At test time, we integrate from the base to obtain $\hat{\mathbf{F}}$, then render actions at the controller's required rate via

$$\hat{\mathbf{A}} = \text{IDCT}_N(\hat{\mathbf{F}}).$$

Because $\hat{\mathbf{F}}$ is rate-agnostic, the same plan can be evaluated at different control frequencies (different $N$) without retraining or changing the policy representation.

## 5 EXPERIMENTS

We evaluate Fourier Flow Matching (FFM) against a stepwise flow-matching baseline (Black et al., 2024) in two stages. First, in simulation, to sanity-check the representation and test whether a policy pretrained on raw Cartesian/joint actions can be finetuned to a Fourier parameterization. Second, on real robots, to assess task performance and compression trade-offs.

The real-world experiments are designed to answer two questions:

1. Does the Fourier representation improve the performance of flow-matching VLA policies?
2. How does the compression rate affect performance?

In all experiments, both models use the same visual encoder, language conditioning, and training protocol; the only difference is the action representation (raw stepwise actions vs DCT coefficients with inverse-DCT reconstruction). We note, however, that both models are finetuned from the policy of Black et al. (2024). This initialization favors the stepwise baseline, because its pretrained representation aligns with the finetuning target, whereas FFM must map pretrained knowledge into coefficient space. Reported FFM gains are therefore conservative with respect to initialization bias.

### 5.1 METRICS

Success rate is the primary metric (reported with Wilson confidence intervals), as ultimately in the robotics context the most important aspect of a policy is the ability to fully complete a given task. We also track completion time as a secondary metric that serves as a proxy of the quality of the performance (faster is better). Since this metric aims to tell us how quickly the model can *complete* the task, it is only considered if the given evaluation was successful.

### 5.2 EXPERIMENTAL ENVIRONMENTS

Figure 3 depicts the experimental environments.

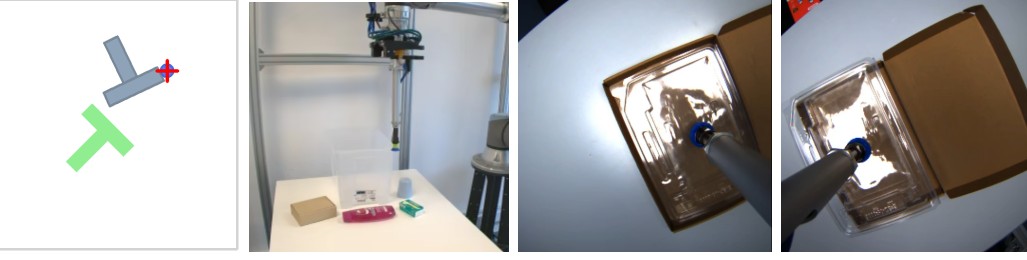

Figure 3: Illustration of the evaluation environments: PushT (top-left), collecting (middle-left), precise-packing (middle-right and right)

### 5.2.1 SIMULATION

We evaluate in the publicly available PushT benchmark (Chi et al., 2024), using the official environment for a controlled comparison between the stepwise flow-matching baseline (Black et al., 2024) and our FFM model. Both policies are run on identical test episodes.

### 5.2.2 REAL-WORLD

We deploy on a UR10 robot with a vacuum (suction) end-effector and consider two tabletop tasks:

- **Collecting:** given a language instruction specifying the target object, pick it and place it into a plastic container. At episode start, four objects and the container are randomly placed on the desk.

- **Precise packing:** a dexterous alignment task in which a transparent plastic wrapper must be inserted into a cardboard box with matching dimensions. Both items start in randomized poses on the table (Figures 2 and 3). The task is intentionally tight-tolerance: even slight rotational or translational misalignment prevents insertion, creating a challenging, practically motivated (packaging/assembly) benchmark where success-rate differences between approaches are readily observable and meaningful.

In both cases we cut off the evaluation at a predefined threshold.

### 5.3 ASYNCHRONOUS INFERENCE

Prior work suggests that overlapping planning with execution improves the reactivity of VLA policies (Black et al., 2025). Our tasks are near-unimodal, so we use a simple pipelined scheme: while executing the current control segment of length $H$, we launch the next inference early so its plan is ready before the segment ends. We discard the prefix corresponding to the actions that were already executed during inference and replace only the remaining suffix. The identical schedule, latency budget, and switch rule are used for both the baseline and FFM policies, isolating the effect of the action representation.

## 6 RESULTS

### 6.1 SUCCESS RATE AND COMPLETION TIME

#### 6.1.1 SIMULATION

We report success rate only, using PushT to check whether a policy pretrained on raw actions can be finetuned to the Fourier parameterization. The baseline attains a slightly higher mean success, but the Wilson 95% confidence intervals overlap, indicating no statistically reliable difference (see Figure 4 and Table 1).

#### 6.1.2 REAL ROBOT

On hardware we evaluate both success rate and completion time (measured only on successful trials). Fourier Flow Matching (FFM) substantially improves success, with a relative gain of $\sim 43\%$ on the most challenging precise packing task (0.715 vs. 0.500). We hypothesize that operating in the Fourier domain helps the policy separate informative high-frequency components from noise. Completion times (Table 2) decrease as well, though the effect is smaller than for success rate.

Table 1: Success rate. Number of evaluations $n$ in brackets

|  | PushT ($n$=190) | Collecting ($n$=345) | Precise packing ($n$=200) |
| --- | --- | --- | --- |
| Baseline | $0.984 \pm 0.020$ | $0.730 \pm 0.047$ | $0.500 \pm 0.069$ |
| FFM (ours) | $0.960 \pm 0.029$ | $0.843 \pm 0.039$ | $0.715 \pm 0.062$ |

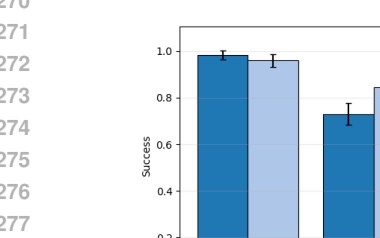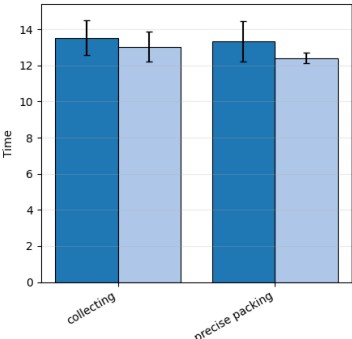

Figure 4: Success rate (left) and mean completion time on successful trials (right). Error bars show 95% CIs; Wilson intervals are used for the binary success metric.

Table 2: Completion time

|  | Collecting | Precise packing |
|---|---|---|
| Baseline | $13.527 \pm 0.973$ | $13.042 \pm 0.808$ |
| FFM (ours) | $12.410 \pm 0.555$ | $12.410 \pm 0.301$ |

## 6.2 IMPACT OF COMPRESSION RATE

We vary the number $K$ of retained DCT-II coefficients to study the compression–performance trade-off. Results are reported on precise packing, our most challenging setting, where effect sizes are easiest to detect statistically. As shown in Figure 5 and Table 3, high success is achieved even with small $K$, indicating that the leading coefficients capture the dominant trajectory structure while attenuating high-frequency noise. In practice, we find a broad plateau of $K$ values that maintain near-peak success, suggesting that $K$ can be selected based on resource constraints without materially affecting performance.

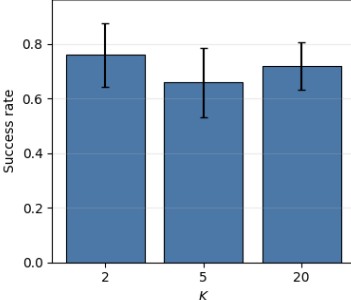

Figure 5: Effect of the number of DCT coefficients $K$ on success rate. Error bars show 95% CIs (Wilson).

Table 3: Impact of the number of DCT coefficients on the success rate.

|  | $K = 2$ ($n = 50$) | $K = 5$ ($n = 50$) | $K = 20$ ($n = 100$) |
|---|---|---|---|
| FFM (ours) | $0.760 \pm 0.116$ | $0.660 \pm 0.127$ | $0.720 \pm 0.086$ |

## 7 SUMMARY AND DISCUSSION

We introduced a Fourier-domain parameterization for VLA policies trained with flow matching, representing trajectories by a small set of DCT coefficients instead of stepwise actions. Across tasks, operating in coefficient space improves success rate and yields modest reductions in completion time. We hypothesize that the representation's compactness and its implicit bias toward smooth, low-frequency structure help the policy separate informative variation from noise while preserving task-critical high-frequency components when needed.

Viewing demonstrations as continuous signals is especially useful for planning and replanning. Flow matching in coefficient space transports distributions over entire trajectories, producing reconstructions that are continuous by construction.

Several directions for future research follow naturally. Decoupling trajectory shape (coefficients) from execution duration is promising: predicting time alongside coefficients may let the policy scale motions to task difficulty without retraining. Alternative compact bases such as B-splines may further improve controllability and robustness. Finally, large-scale pretraining with trajectories converted to the Fourier domain, followed by task-specific finetuning, could test whether the observed benefits persist at scale.

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
