# OpenReview forum: "Shaping Robotic Actions with Fourier Flow Matching"
_ICLR.cc/2026/Conference — ICLR 2026 Conference Withdrawn Submission_

### Official Review · Reviewer_Q3yZ · 2025-10-27

**Soundness:** 2
**Presentation:** 1
**Contribution:** 1
**Rating:** 2
**Confidence:** 3

**Summary:**

This paper proposes a Fourier-based flow-matching method (FFM) for VLA policies, which is presented as a variant of flow matching. Compared to standard flow matching, FFM performs flow matching in a compact Fourier domain using forward and inverse Discrete Cosine Transform (DCT). The authors argue that this trajectory-level representation yields smoother and lower-dimensional embeddings that facilitate policy learning. However, the overall contribution is thin: the work primarily grafts a common signal processing tool (DCT) into flow matching. With very limited experiments, the paper fails to convincingly validate its claims or provide new insights.

**Strengths:**

The choice of representation space is indeed a relevant issue for VLA research. The DCT-based representation seems simple, plug-and-play, and easy to implement.

**Weaknesses:**

1.	Limited contribution: The method is essentially just a change of representation (stepwise actions → DCT space). The properties of this space are not theoretically or empirically validated. Moreover, the paper does not clarify what practical advantages this representation enables (e.g., can it support cross-embodiment policies?).

2.	Lack of theoretical analysis: The paper provides no rigorous theoretical justification. The claim that “DCT enforces smoothness and reduces noise” is neither new nor well supported.

3.	Insufficient experimental support: Experiments are restricted to PushT simulation and two simple tabletop tasks (collecting and precise packing) on UR10. These toy tasks are insufficient to demonstrate scalability. In PushT, the baseline even slightly outperforms FFM; only in packing does FFM achieve a relative gain. The study compares against only one baseline (stepwise flow matching, Black et al. 2024) and omits broader comparisons (e.g., diffusion policy, autoregressive or tokenized action methods).

4.	Poor literature coverage: The paper cites only 11 references, which fail to reflect the breadth of prior work. Many relevant works on trajectory/action representations and policy learning are not cited.

**Questions:**

In addition to the weaknesses above:

1.	Can the authors provide more theoretical analysis to substantiate claims of smoothness and explain the practical downstream benefits of this representation (e.g., improved downstream success rates for pre-training VLAs, cross-embodiment generalization)?

2.	Can the authors compare against stronger baselines, such as diffusion policies, autoregressive models, or tokenization-based approaches?

3.	Can FFM be scaled to large-scale VLA training or cross-embodiment transfer, which are central to current VLA research?

4.	As a flow matching variant, can FFM improve the performance of open-source foundation VLA models (e.g., Pi0.5, GR00t)?

5.	Since DCT coefficient count K is not linearly correlated with success rate, does one have to rely on hyperparameter search for tuning?

6.	How does FFM perform in contact-rich tasks or fine-grained dexterous hand manipulation scenarios?

---

### Official Review · Reviewer_F1vP · 2025-10-31

**Soundness:** 2
**Presentation:** 1
**Contribution:** 2
**Rating:** 0
**Confidence:** 4

**Summary:**

This paper proposes to improve on robot control actions generated from VLAs by learning to generate the fourier coefficients of the flow field, rather than the direct flow field itself. Generating fourier coefficients of the flow field helps to ensure smooth outputs and also helps to reduce the dimensionality of the generation process.

**Strengths:**

The overall idea is interesting and reasonable. The proposed modification to flow generation is relatively straight forward.

**Weaknesses:**

Unfortunately, the paper leaves a great many questions unanswered. The overall idea is not well-known but is also not completely novel, in that there have been some recent papers in functional flow matching (including Fourier functional bases), such as Kerrigan et al 2023 and Li et al 2024. How does this paper relate to that work in functional flow matching? The related work section and references are relatively brief.

The technical development is even more cursory, as there is not description of the training process, but simply a statement of what the DCT is and how the DCT co-efficients can be used to generate a flow field. There is no discussion of the training data sampling, and the relationship to the frequencies of the corresponding flow field.

The only reported metrics are success rate and completion time, but the purpose of the paper is to produce better (e.g., smoother) trajectories. It is interesting that on PushT, the baseline does slightly better (if not statistically significantly better) in success rate and the authors do not report completion time for this task. The differences between the policies are not substantial, which raises questions about whether or not this technique is especially effective in any way.

There is some analysis of the compression achieved, and one of the promises is to reduce the dimensionality, but there is no comparison to a time-series flow field representation.

The experimental demonstrations on the robot are relatively uncompelling. There is no description of a trajectory that is poorly generated by the baseline time-series flow field, and an improved trajectory from the Fourier flow field.

**Questions:**

How do the authors propose to train the model?

How stable are the Fourier flow fields?

How do the authors propose to address sampling questions?

---

### Official Review · Reviewer_k8dz · 2025-11-05

**Soundness:** 3
**Presentation:** 3
**Contribution:** 2
**Rating:** 4
**Confidence:** 3

**Summary:**

The work proposes Fourier Flow Matching (FFM) for VLA robot policies. Instead of predicting step-wise joint/Cartesian actions like existing flow-matching models (e.g., pi-0), the FFM predicts Discrete Cosine Transform (DCT) coefficients of the trajectory and reconstructs the action sequence by inverse DCT. This encourages smoothness and reduces the dimensionality of the action. Experimental results show similar performance to a step-wise baseline in simulation and improvements on two real-world robot tasks.

**Strengths:**

* The motivation is clear, step-wise flows introduce discontinuities and high-frequency noise.
* The methodology/implementation is straightforward, i.e., no need for tokenizer training, drop-in usage of DCT.
* Real hardware deployments: showing improvements in precision tasks.
* Good ablation on the number of coefficients K.

**Weaknesses:**

* The originality is somehow incremental
	* The combination of DCT and flow-based VLA is expected since prior works (e.g., FAST/BEAST) already introduced Fourier/spline action compression effectiveness.
* Evaluations are limited
	* Only a single baseline is considered
	* Lacks of comparions to strong AR + tokenizer models (e.g., FAST)
* The robustness/generality of the work is not thoroughly discussed
	* Only show improvements for tabletop tasks, no other types of manipulations
* Insufficient discussion of the selection of K
	* How to tune K in an empirical way?

**Questions:**

* Did author(s) try to train FFM from scratch? How would it end?
* Why no comparison to FAST/BEAST, as they also adopt similar approaches with strong performance?
* Could the author(s) provide some principles that help to choose K since the results are nonmonotonic?
* What is the inference latency w.r.t. existing flow-based VLA?

---

### Official Review · Reviewer_j6VK · 2025-11-08

**Soundness:** 1
**Presentation:** 1
**Contribution:** 1
**Rating:** 0
**Confidence:** 5

**Summary:**

This paper introduces Fourier Flow Matching (FFM), an approach that replaces stepwise joint/Cartesian actions in Vision–Language–Action (VLA) flow-matching policies with a compact set of DCT-II coefficients. The model learns a probability flow directly in coefficient space and reconstructs actions via inverse DCT at inference time. The authors argue that this parameterization enforces smoothness, reduces dimensionality, and decouples planning from controller rate. Experiments on PushT and two real-robot tasks show moderate improvements in success rate and slight reductions in completion time compared to a stepwise flow-matching baseline.

**Strengths:**

1. The paper introduces a simple, elegant modification to flow-matching VLA policies: representing actions in the Fourier domain via DCT coefficients. This idea is not new, but well-motivated, and it leverages established signal-processing principles to enforce smoothness and reduce high-frequency noise.

2. The real-robot results—particularly on precise packing—show substantial gains in task success (e.g., 0.715 vs 0.500).

**Weaknesses:**

1. While the application of DCT parameterization to flow matching is new, action compression through Fourier[1], spline[2], or movement primitives[3] representations is not. The paper’s methodological novelty is therefore incremental: essentially swapping the raw action representation for DCT coefficients.

2. The paper heavily motivates the DCT choice through intuition (smoothness priors, low-frequency dominance) but does not offer theoretical justification for why DCT is explicitly well matched to robot motion in VLA settings compared to other bases (splines, RVQ[4], movement primitives, etc.).

3. No comparison against non-flow-matching, trajectory-based models. Recent autoregressive VLA models using learned tokenizers (e.g., FAST[1], BEAST[2]) could serve as competitive baselines.

4. The empirical study is inefficient. The method is evaluated with only a single VLA backbone ($\pi_0$), and tested in one simulation environment (PushT) and two relatively simple real-robot tasks. This makes it difficult to assess robustness, generalizability, or task diversity. Evaluation on additional benchmark suites—such as LIBERO or CALVIN, which contain multi-stage, long-horizon, visually diverse manipulation tasks—would significantly strengthen the empirical claims.

5. The writing quality requires substantial improvement. The paper currently reads more like a technical report than a polished conference submission.

[1] Pertsch, Karl, et al. "Fast: Efficient action tokenization for vision-language-action models." RSS 2025.

[2] Zhou, Hongyi, et al. "BEAST: Efficient Tokenization of B-Splines Encoded Action Sequences for Imitation Learning." arXiv preprint arXiv:2506.06072 (2025).

[3] Scheikl, Paul Maria, et al. "Movement primitive diffusion: Learning gentle robotic manipulation of deformable objects." IEEE Robotics and Automation Letters 9.6 (2024): 5338-5345.

[4] Lee, Seungjae, et al. "Behavior generation with latent actions." ICML 2024.

**Questions:**

See weakness

---

### Note · Authors · 2025-11-28

**Comment:**

We agree with the reviewers that the submission requires additional work, which we believe goes beyond the scope of the rebuttal process.

**Withdrawal Confirmation:**

I have read and agree with the venue's withdrawal policy on behalf of myself and my co-authors.